# Using Machine Learning for the Calibration of Airborne Particulate Sensors

**DOI:** 10.3390/s20010099

**Published:** 2019-12-23

**Authors:** Lakitha O.H. Wijeratne, Daniel R. Kiv, Adam R. Aker, Shawhin Talebi, David J. Lary

**Affiliations:** University of Texas at Dallas, 800 W, Campbell Rd, Richardson, TX 75080, USA; drk150030@utdallas.edu (D.R.K.); Adam.Aker@utdallas.edu (A.R.A.); Shawhin.Talebi@utdallas.edu (S.T.); David.Lary@utdallas.edu (D.J.L.)

**Keywords:** optical particle counter, airborne particulates, machine learning

## Abstract

Airborne particulates are of particular significance for their human health impacts and their roles in both atmospheric radiative transfer and atmospheric chemistry. Observations of airborne particulates are typically made by environmental agencies using rather expensive instruments. Due to the expense of the instruments usually used by environment agencies, the number of sensors that can be deployed is limited. In this study we show that machine learning can be used to effectively calibrate lower cost optical particle counters. For this calibration it is critical that measurements of the atmospheric pressure, humidity, and temperature are also made.

## 1. Introduction

Airborne atmospheric aerosols are an assortment of solid or liquid particles suspended in air [1]. Aerosols, also referred to as particulate matter (PM), are associated with a suite of issues relevant to the global environment [2,3,4,5,6,7,8], atmospheric photolysis, and a range of adverse health effects [9,10,11,12,13,14,15]. Atmospheric aerosols are usually formed either by direct emission from a specific source (e.g., combustion) or from gaseous precursors [16]. Although individual aerosols are typically invisible to the naked eye, due to their small size, their presence in the atmosphere in substantial quantities means that their presence is usually visible as fog, mist, haze, smoke, dust plumes, etc. [17]. Airborne aerosols vary in size, composition, origin, and spatial and temporal distributions [14,18]. As a result, the study of atmospheric aerosols has numerous challenges.

### 1.1. Motivation for This Study

Low cost sensors that can also be accurately calibrated are of particular value. For the last two decades we have pioneered the use of machine learning to cross-calibrate sensors of all kinds. This was initially done for very expensive orbital instruments onboard satellites (awarded an IEEE paper prize, and specially commended by the NASA MODIS team) [19]. We are now using this approach operationally for low-cost sensors distributed at scale across dense urban environments as part of our smart city sentinels. The approach can be used for very diverse sensors, but as a useful illustrative example that has operational utility, we describe here a case for accurately calibrated, low-cost sensors measuring the abundance and size distribution of airborne particulates, with the implicit understanding that many other sensor types could easily be substituted. These sensors can be readily deployed at scale at fixed locations; be mobile on various robotic platforms (walking, flying, etc) or vehicles; be carried; or deployed autonomously as a mesh network, either by operatives or by robots (walking, flying, etc.).

Building in calibration will enable consistent data to retrieved from all the low-cost nodes deployed/thrown. Otherwise the data will always be under some suspicion as the inter-sensor variability among low-cost nodes can be substantial. While much effort has been recently placed on providing the connectivity of large disbursed low-cost networks, little to no effort has been spent on the automated calibration, bias-detection, and uncertainty estimation necessary to make sure the information collected is sound. A case study of providing this critical calibration using machine learning is the focus of this paper.

Any sensor system benefits from calibration, but low-cost sensors are typically in particular need of calibration. The inter-sensor variability among low-cost nodes can be substantial. In addition, to the pre-deployment calibration, once the sensors have been deployed, the paradigm we first developed for satellite validation of constructing probability distribution functions of each sensor’s observation streams, can be used to both monitor the real-time calibration of each sensor in the network by comparing its readings to those of its neighbors, but also to answer the question “how representative is an instantaneous reading of the conditions seen over some temporal and spatial window within which the sensor is placed?”.

### 1.2. Using Probability Distribution Functions to Monitor Calibration and Representativeness in Real-Time

It is useful to be able to answer the question, “How representative is an instantaneous reading of the conditions seen over some temporal and spatial window within which the sensor is placed?”. We can answer this question by considering a probability distribution functions (PDFs) of all the observations made by a sensor over some temporal and spatial window [20]. The width of this probability distribution is termed the representativeness uncertainty for that temporal and spatial window. The PDFs of all observations made by each sensor are automatically compared in real time to the PDFs from the neighboring sensors within a neighborhood radius. These neighborhood sensors can include measurements from primary reference sensors that may be available. This comparison is used to estimate the measurement uncertainty and inter-instrument bias for the last hour, day, etc. We continuously accumulate the PDFs for each sensor over a variety of time scales and compare it to its nearest neighbors within a neighborhood radius. Any calibration drift in a sensor will be quickly identified as part of the fully automated, real-time workflow, wherein we will automatically be comparing each sensor’s PDFs to its neighbor’s PDFs, and to the reference instrument’s PDFs. As each sensor is in a slightly different local environment, the sensor bias drift for each sensor will be different.

### 1.3. Characterizing the Temporal and Spatial Scales of Urban Air Pollution

This study focused on the calibration of low cost sensors is part of a larger endeavor with the goal of characterizing the temporal and spatial scales of urban pollution. The temporal and spatial scales of each atmospheric component are intimately connected. The resolution used in atmospheric chemistry modeling tools is often driven by the computational resources available. The spatial resolution of observational networks is often determined by the fiscal resources available. It is worth taking a step back and characterizing what the actual spatial scales are for each chemical component of urban atmospheric chemistry. Based on our street level surveys providing data at resolution higher than one meter, it is clear that the spatial scales are dependent on several factors—the synoptic situation, the distribution of sources, the terrain, etc. In the larger study we characterized the spatial scales of multi-specie urban pollution by using a hierarchy of measurement capabilities that include: (1) A zero emission electric survey vehicle with comprehensive gas, particulate, irradiance, and ionizing radiation sensing. (2) An ensemble of more than one hundred street level sensors making measurements every few seconds of a variety of gases, and of particulates, light levels, temperature, pressure, and humidity. Each sensor is accurately calibrated against a reference standard using machine learning. This paper documents an example of low-cost sensor calibration for airborne particulate observations.

### 1.4. Societal Relevance

What are the characteristic spatial scales of each chemical species and how does this depend on issues such as the synoptic situation? These are basic questions that are helpful to quantify when considering atmospheric chemistry; when looking forward to the next generation of modeling tools and observing systems (whether from space or ground-based networks); and when evaluating mitigation strategies, especially with regard to co-benefits for air pollution and greenhouse gas reduction and investigating the evolution of urban air composition in a warming climate. To be able to quantify these spatial and temporal scales we need a comprehensive observing system, so being able to use low cost sensors is of great assistance to achieving this goal.

The Dallas–Fort Worth (DFW) metroplex (where our study was conducted) is the largest inland urban area in the United States and the nation’s fourth largest metropolitan area. Nearly a third of Texans, more than seven million inhabitants, live in the DFW area. A population which is growing by a thousand people every day. DFW is an area with an interesting variety of specific pollution sources with unique signatures that can provide a useful testbed for generalizing a measurement strategy for dense urban environments. For more than two decades the DFW area has been in continuous violation of the Clean Air Act. DFW will be one of only ten non-California metropolitan areas still in violation of the Clean Air Act in 2025 unless major changes take place. This has already had a detrimental health impact; e.g., even though the average childhood asthma rate is 7% in Texas, and the national average is 9%, the DFW childhood asthma rate is 20%–25%. Second only to the Northeast, DFW ranks second in the number of annual deaths due to smog. Further, a leading factor in poor learning outcomes in high-schools is absenteeism, a leading cause of absenteeism is asthma, and key trigger for asthma is airborne pollution [21]. Physical exertion in the presence of high pollution levels is more likely to lead to an asthmatic event. The sensors calibrated in this study were provided to high schools and high school coaches so that simple, practical decisions can be made to reduce adverse health outcomes; e.g., given the levels of pollen/pollution today, should physical education/practice be outside or inside?

## 2. The Datasets Used

All of the measurements were made at our own field calibration station in an ambient environment. The calibration of the low cost AphaSense OPC occurred prior to their deployment across the dense urban environment of DFW. In this study we used machine learning to bring together two distinct types of data. First, we used accurate in-situ observations made by a research grade particulate spectrometer. Second, we used observations from inexpensive optical particle counters. The inexpensive sensors are particularly useful as they can be readily deployed at scale.

### 2.1. Research Grade Optical Particle Counter

The particulate spectrometer is a laser based Optical Particle Counter (OPC). In this study we used a GRIMM Laser Aerosol Spectrometer and Dust Monitor Model 1.109 (Germany). The sensor has the capability of measuring particulates of diameters between 0.25 and 32 μm distributed within 32 size channels. Such a wide range of diameter space is made possible due to intensity modulation of the laser source. Particulates pumped into the sensor are detected through scattering a laser beam of 655 nm into a light trap. The laser beam is aimed at particulates coming through a sensing chamber at a flow-rate of 1.21 L/min. The device classifies particulates into specific size classes subject to its intensity [22]. The optical arrangement of the sensor is staged such that a curved optical mirror placed at an average scattering angle of 90${}^{\circ }$ collects and redirects the scattered light towards a photo sensor. The wide angle of the optical mirror (120${}^{\circ }$) is meant to increase the light intensity redirected towards the photo sensor within the Rayleigh scattering domain which decreases the minimum detectable particle size. Furthermore, it compensates for Mie Scattering undulations caused by monochromatic illumination. The sensing period of the GRIMM sensor was set to 6 s, and for each time window provided three standardized mass fractions; namely, based on occupational health (repairable, thoracic, and alveolic) according to EN 481, and PM${}_{1}$, PM${}_{2.5}$, and PM${}_{10}$.

### 2.2. Low Cost Optical Particle Counters

There are several readily available optical particle counters (OPC) which are useful, but much less accurate compared to research grade sensors. In this study, we focus on using such sensors, together with machine learning, to get as close as possible to the accuracy of research grade PM sensors. After the application of the machine learning calibration, these lower cost sensors perform admirably. In order for low cost sensors to provide an improved picture of PM levels, a careful calibration is required. The current study used an Alpha Sense OPC-N3 (http://www.alphasense.com/) together with a cheaper environmental sensor (Bosch BME280) as data collectors. The OPC-N3 is compact (75 mm × 60 mm × 65 mm) in size and weighs under 105 g, but uses similar technology to the conventional OPCs where particle size is determined via a calibration based on Mie scattering. Unlike most OPCs the OPC-N3 does not include a pump and a replaceable particle filter in order to pump aerosol samples through a narrow inlet tube; hence, avoiding the need for regular maintenance. A sufficient airflow through the sensor is made possible with a low powered micro fan producing a sample flow rate of 280 mL/min. The OPC-N3 is capable of on-board data logging and measuring particulates with diameters up to 40 μm. This enables the OPC-N3 to measure pollen and other biological particulates. The on-board data is saved within an SD card which can be accessed through micro-USB cable connected to the OPC. Furthermore, the OPC-N3’s lower sensing diameter is 0.35 μm, as opposed to its predecessor’s (OPC-N2) limit of 0.38 μm. The wider range of sensing is made possible via the OPC switching between high and low gain modes automatically. The OPC-N3 calculates its PM values using the method defined by the European Standard EN 481 [23].

### 2.3. Caveat: Particulate Refractive Index

The observations made by optical particle counters are sensitive to the refractive index of the particulates and their light absorbing properties. The retrieved size distributions and the mass-concentrations can be biased, depending on the nature of the particulates. The current study did not explore the accuracy implications of this. A future study is underway which includes direct measurements of black carbon that will allow us to begin to explore these aspects. The machine learning paradigm is readily extensible to include these aspects, even though not explicitly addressed in this study. Machine learning is an ideal approach for the calibration of lower cost optical particle counters.

## 3. Machine Learning

Machine learning has already proved useful in a wide variety of applications in science, business, health care, and engineering. Machine learning allows us to *learn by example*, and to *give our data a voice*. It is particularly useful for those applications for which we do *not* have a complete theory, yet which are of significance. Machine learning is an automated implementation of the scientific method [24], following the same process of generating, testing, and discarding or refining hypotheses. While a scientist or engineer may spend their entire career coming up with and testing a few hundred hypotheses, a machine-learning system can do the same in a fraction of a second. Machine learning provides an objective set of tools for automating discovery. It is therefore not surprising that machine learning is currently revolutionizing many areas of science, technology, business, and medicine [25,26].

Machine learning is now being routinely used to work with large volumes of data in a variety of formats, such as images, videos, sensors, health records, etc. Machine learning can be used in understanding this data and create predictive and classification tools. When machine learning is used for regression, empirical models are built to predict continuous data, facilitating the prediction of future data points, e.g., algorithmic trading and electricity load forecasting. When machine learning is used for classification, empirical models are built to classify the data into different categories, aiding in the more accurate analysis and visualization of the data. Applications of classification include facial recognition, credit scoring, and cancer detection. When machine learning is used for clustering, or unsupervised classification, it aids in finding the natural groupings and patterns in data. Applications of clustering include medical imaging, object recognition, and pattern mining. Object recognition is a process for identifying a specific object in a digital image or video. Object recognition algorithms rely on matching, learning, or pattern recognition algorithms using appearance-based or feature-based techniques. These technologies are being used for applications such as driver-less cars, automated skin cancer detection, etc.

Machine learning is an automated approach to building empirical models from the data alone. A key advantage of this is that we make no a priori assumptions about the data, its functional form, or its probability distributions. It is an empirical approach. However, it also means that for machine learning to provide the best performance we do need a comprehensive, representative set of examples, that spans as much of the parameter space as possible. This comprehensive set of examples is referred to as the ‘training data’.

So, for a successful application of machine learning we have two key ingredients, both of which are essential, a machine learning algorithm, and a comprehensive training data set. Then, once the training has been performed, we should test its efficacy using an independent validation data set to see how well it performs when presented with data that the algorithm has not previously seen; i.e., test its ‘generalization’. This can be, for example, a randomly selected subset of the training data that was held back and then utilized for independent validation.

It should be noted, that with a given machine learning algorithm, the performance can go from poor to outstanding with the provision of a progressively more complete training data set. Machine learning really is learning by example, so it is critical to provide as complete a training data set as possible. At times, this can be a labor intensive endeavor.

We have used machine learning in many previous studies [19,21,25,26,27,28,29,30,31,32,33,34,35,36,37,38,39,40,41,42,43,44,45,46,47,48,49,50,51,52,53,54,55,56]. In this study we used machine learning for multivariate non-linear non-parametric regression. Some of the commonly used regression algorithms include neural networks [57,58,59,60,61,62], support vector machines [63,64,65,66,67], decision trees [68], and ensembles of trees such as random forests [69,70,71]. Previously we used a similar approach to cross-calibrate satellite instruments [19,25,26,27,28]. Recently other studies also used machine learning to calibrate low cost sensors [72,73].

### Ensemble Machine Learning

Multiple approaches for non-linear non-parametric machine learning were tried, including neural networks, support vector regression, and ensembles of decision trees. The best performance was found using an ensemble of decision trees with hyper-parameter optimization [68,69,70,71]. The specific implementation used was that provided by the Mathworks in the fitrensemble function which is part of the Matlab Statistics and Machine Learning Toolbox. Hyperparameter optimization was used so that the optimal choice was made for the following attributes: learning method (bagging or boosting), maximum number of learning cycles, learning rate, minimum leaf size, maximum number of splits, and the number of variables to sample.

There were 72 inputs to our multivariate non-linear non-parametric machine learning regression; these included the particle counts for each of the 24 size bins measured by the OPC-N3; the OPC-N3 estimates of PM${}_{1}$, PM${}_{2.5}$, and PM${}_{10}$; a suite of OPC performance variables, including the reject ratio; and particularly importantly, the ambient atmospheric pressure, temperature, and humidity. The OPC-N3 sensor includes two photo diodes that record voltages which are eventually translated into particle count data. However, particles which are not entirely in the OPC-N3 laser beam, or are passing down the edge, are rejected and this is recorded in the “reject ratio” parameter. This leads to better sizing of particles, and hence plays an important role within the machine learning calibration.

Each of the six outputs we wished to estimate had its own empirical model. The performances of each of these six models in their independent validations are shown in Figure 1 and Figure 2. The outputs we estimated were the six variables measured by the reference instrument, the research grade optical particle counts, namely, of PM${}_{1}$, PM${}_{2.5}$, and PM${}_{10}$; and the standardized occupational health repairable, thoracic, and alveolic mass fractions. The alveolic fraction is the mass fraction of inhaled particles penetrating to the alveolar region (maximum deposition of particles with a size ≈2 μm). The Thoracic fraction is the mass fraction of inhaled particles penetrating beyond the larynx (<10 μm). The respirable fraction is the mass fraction of inhaled particles penetrating to the unciliated airways (<4 μm). The inhalable fraction is the mass fraction of total airborne particles which is inhaled through the nose and mouth (<20 μm). For each of these six parameters we created an empirical multivariate non-linear non-parametric machine learning regression model with hyper-parameter optimization.

## 4. Results

### Calibrating the Low Cost Optical Particle Counters Using Machine Learning

Figure 1 shows the the results of the multivariate non-linear non-parametric machine learning regression for PM${}_{1}$ (panels a to c), PM${}_{2.5}$ (panels d to f), and PM${}_{10}$ (panels g to i). The left hand column of plots shows the log–log axis scatter diagrams with the *x*-axis showing the PM abundance from the expensive reference instrument and the y-axis showing the PM abundance provided by calibrating the low-cost instrument using machine learning.

For the left hand column of plots in Figure 1 (the scatter diagrams), for a perfect calibration, the scatter plot would be a straight line with a slope of one and a y-axis intercept of zero; the blue line shows the ideal response. We can see that multivariate non-linear non-parametric machine learning regression that we used in this study employing an ensemble of decision trees with hyper-parameter optimization performed very well (panels a, d, and g). In each scatter diagram the green circles are the data used to train the ensemble of decision trees; the red pluses are the independent validation data used to test the generalization of the machine learning model.

We can see that the performance is best for the smaller particles that stay lofted in the air for a long period and do not rapidly sediment, so when comparing the scatter diagram correlation coefficients, *r*, for the independent validation test data (red-points) we see that $r_{{\mathrm{PM}}_{1}}$ > $r_{{\mathrm{PM}}_{2.5}}$ > $r_{{\mathrm{PM}}_{10}}$.

For the middle column of plots in Figure 1 (the quantile–quantile plots), we are comparing the *shape* of the probability distribution (PDF) of all the PM abundance data collected by the expensive reference instrument to that of the the PM abundance provided by calibrating the low-cost instrument using machine learning. A log${}_{10}$ scale is used with a tick mark every decade. The dotted red line in each quantile–quantile plot shows the ideal response. The red numbers indicate the percentiles (0, 25, 50, 75, 100). If the quantile–quantile plot is a straight line, that means both PDFs have *exactly* the same shape, as we are plotting the percentiles of one PDF against the percentiles of the other PDF. Usually, we would like to see a straight line at least between the 25th and 75th percentiles; in this case, we have a straight line over the entire PDF, which demonstrates that the machine learning calibration performed well.

The right hand column of plots shows the relative importance of the input variables for calibrating the low-cost optical particle counters using machine learning. The relative importance metric is a measure of the error that results if that input variable is omitted. In the right hand column of bar plots we have sorted the importance metrics into descending order, so the variable represented by the uppermost bar in each each case was the most important variable for performing the calibration; the second bar was the second most important; etc. We note that along with the number of particles counted in each size bin, it is important to measure the temperature, pressure, and humidity to be able to accurately calibrate the low cost OPC against the reference instrument. The data also suggests that the parameter “reject ratio” carries a greater deal of importance with respect to the calibration. OPC-N3 comprises two photo diodes which records voltages which are eventually translated into particle count data. However, particles which are not entirely in the beam or are passing down the edge are rejected and that is reflected on the parameter “reject ratio”. This leads to better sizing of particles, and hence plays a vital role within the ML calibration.

Another division of occupational health based size-selective sampling is defined by assessing the subset of particles that can reach a selective region of the respiratory system. On this basis three main fractions were defined: inhalable, thoracic, and respirable [74,75,76]. Studies have shown that exposure of excess particulate matter has alarming negative health effects [77]. The smallest sizes of particulate matter are capable of penetrating through to the lungs or even to one’s blood stream.

Figure 2 is similar to Figure 1 and shows the results of the multivariate non-linear non-parametric machine learning regression for the alveolic, thoracic, and inhalable size fractions. As would be expected, we see that the performance is best for the smaller particles that stay lofted in the air for a long period and do not rapidly sediment, so when comparing the scatter diagram correlation coefficients, *r*, for the independent validation test data (red-points) we see that $r_{Alveolic}$ > $r_{Thoracic}$ > $r_{Inhalable}$.

## 5. Operational Use of the Calibration and Periodic Validation Updates

The calibration just described occurred pre-deployment of the sensors into the dense urban environment. Once these initial field calibration measurements were made over a period of several months, in the manner described above, the multi-variate non-linear non-parametric empirical machine learning model was applied in real time to the live stream of observations coming from each of our air quality sensors deployed across the dense urban environment of the Dallas Fort Worth metroplex. These corrected measurements were then made publicly available as open data and depicted on a live map and dashboard.

Building in continual calibration to a network of sensors will enable long-term, consistent, and reliable data. While much effort has been recently placed on the connectivity of large disbursed IoT networks, little to no effort has been spent on the automated calibration, bias-detection, and uncertainty estimation necessary to make sure the information collected is sound. This is one of our primary goals. This is based on extensive previous work funded by NASA for satellite validation.

After deployment, a zero emission electric car carrying our reference was used, to routinely drive past all the deployed sensors to provide ongoing routine calibration and validation. An electric vehicle does not contribute any ambient emissions, and so, is an ideal mobile platform for our reference instruments.

For optimal performance, the implementation combines edge and cloud computing. Each sensor node takes a measurement at least every 10 s. The observations are continually time-stamped at the nodes and streamed to our cloud server, the central server aggregating all the data from the nodes, and managing them. To prevent data loss, the sensor nodes store any values that have not been transmitted to the cloud server for reasons, including communication interruptions, in a persistent buffer. The local buffer is emptied to the cloud server at the next available opportunity.

Data from all sensors are archived and serve as an open dataset that can be publicly accessed. The observed probability distribution functions (PDFs) from each sensor are automatically compared in real time to the PDFs from the neighboring sensors within a neighborhood radius. These neighborhood sensors include measurements from the electric car/mobile validation sensors. This comparison was used to estimate the size resolved measurement uncertainty and size resolved inter-instrument bias for the last hour, day, week, month, and year. We continuously accumulated the PDF for each sensor over a variety of time scales (h, day, week, month, and year) and compare it to its nearest neighbors within a neighborhood radius.

Any calibration drift in a sensor will be quickly identified as part of a fully automated real-time workflow, where we will automatically be comparing each sensor’s PDFs to its neighbor’s PDFs, and to the reference instruments’ PDFs. As each sensor is in a slightly different local environment, the sensor bias drift for each sensor will be different. We have previously shown that machine learning can be used to effectively correct these inter-sensor biases [19]. As a result, the overall distributed sensing system will not just be better characterized in terms of its uncertainty and bias, but provide improved measurement stability over time.

## 6. Conclusions

We have shown that machine learning can be used to effectively calibrate lower cost optical particle counters. For this calibration it is critical that measurements of the atmospheric pressure, humidity, and temperature are included. Once the machine learning calibration was applied to the low cost sensors, independent validation using scatter diagrams and quantile–quantile plots showed that, not only was the calibration effective, but the shape of the resulting probability distribution of observations was very well preserved.

These low cost sensors are being deployed at scale across the dense urban environment of the Dallas Fort Worth metroplex for characterizing both the temporal and spatial scales of urban air pollution and for providing high schools and high school coaches a tool to assist in making better decisions to reduce adverse health outcomes; e.g., given the levels of pollen/pollution today should physical education/practice be outside or inside?

## Figures and Tables

**Figure 1 sensors-20-00099-f001:**
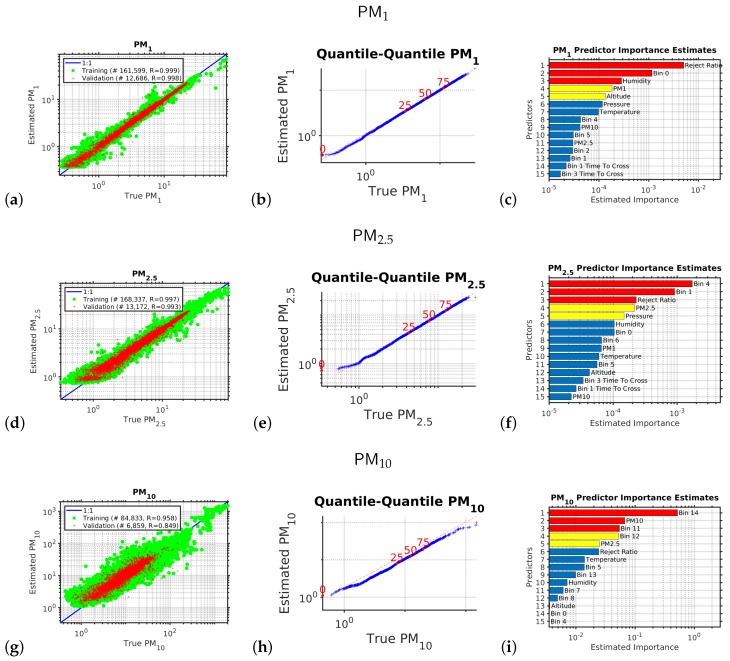
This figure shows the results of the multivariate non-linear non-parametric machine learning regression for particulate matter PM${}_{1}$ (panels (**a**)–(**c**)), PM${}_{2.5}$ (panels (**d**)–(**f**)), and PM${}_{10}$ (panels (**g**)–(**i**)). The left hand column of plots shows the log–log axis scatter diagrams with the x-axis showing the PM abundance from the expensive reference instrument and the y-axis showing the PM abundance provided by calibrating the low-cost instrument using machine learning. The green circles are the training data; the red pluses are the independent validation dataset. The blue line shows the ideal response. The middle column of plots shows the quantile–quantile plots for the machine learning validation data, with the x-axis showing the percentiles from the probability distribution function of the PM abundance from the expensive reference instrument and the y-axis showing the percentiles from the probability distribution function of the estimated PM abundance provided by calibrating the low-cost instrument using machine learning. The dotted red line shows the ideal response. The right hand column of plots shows the relative importance of the input variables for calibrating the low-cost optical particle counters using machine learning.

**Figure 2 sensors-20-00099-f002:**
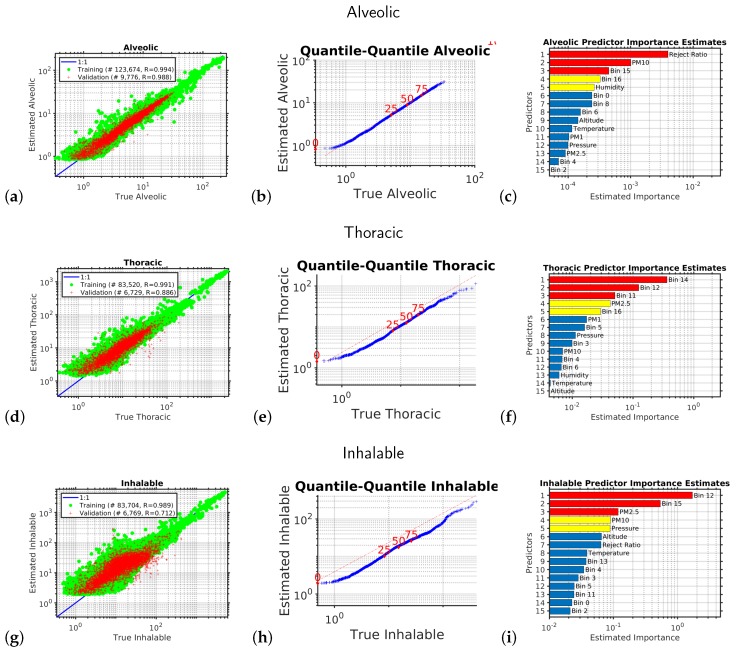
This figure shows the results of the multivariate non-linear non-parametric machine learning regression for the alveolic (panels (**a**)–(**c**)), thoracic (panels (**d**)–(**f**)), and inhalable size fractions (panels (**g**–**i**)). The left hand column of plots shows the log–log axis scatter diagrams with the x-axis showing the PM abundance from the expensive reference instrument and the y-axis showing the PM abundance provided by calibrating the low-cost instrument using machine learning. The green circles are the training data; the red pluses are the independent validation dataset. The blue line shows the ideal response. The middle column of plots shows the quantile–quantile plots for the machine learning validation data, with the x-axis showing the percentiles from the probability distribution function of the PM abundance from the expensive reference instrument and the y-axis showing the percentiles from the probability distribution function of the estimated PM abundance provided by calibrating the low-cost instrument using machine learning. The dotted red line shows the ideal response. The right hand column of plots shows the relative importance of the input variables for calibrating the low-cost optical particle counters using machine learning.

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
