# Peer review of "Using Machine Learning for the Calibration of Airborne Particulate Sensors"

_sensors, 2019, doi:10.3390/s20010099_

Round 1

Reviewer 1 Report

I found the manuscript can be publsihed at it is.

I don't have further comments or suggestions to the authors

Author Response

Thank you for taking the time to read our manuscript. We really appreciate it!

Reviewer 2 Report

This short paper provides a nice study on machine learning to calibrate a low-cost aerosols optical counter. The paper is well written and easy to read. Nevertheless, the paper is too short and some important information are missing. I am sure that such information can be provided by the authors in the revised version of the paper.

Major comments:

My main concern is the use of the Gimm OPC as a reference instrument. Like most of the OPC, it is sensitive to the refractive index of the particulates, and thus to their light absorbing properties. The retrieved size distributions and the mass-concentrations can be biased, depending on the nature of the particulates. Have you re-calibrated the Grimm instrument for ambient dust, or for carbonaceous particles? It is the same problem for the Alpha Sensor OPC. Then, which nature of particulates have you used during your measurements, and how can you consider their impact on the measurement’s accuracy?

The authors must provide the origin of their data. Are they coming from laboratory measurements, from air quality network or from field campaign?

The conclusion is too short. Perhaps the authors could provide some perspective of their work to future studies on ambient air monitoring, field campaign, pollution events …

Finally, we don’t see how this work can be used to calibrate the low-cost sensors. At present, it provides an estimate of their accuracy. The authors must explain how their wok can be applied for such calibration work.

Minor comments:

Line 28: Can you explain « Intensity modulation of the laser source”?

Line 33: Replace “90” by “90°”.

Line 34: Replace “120” by “120°”.

Line 111: Can you define “standardized occupational health Repairable, Thoracic and Alveolic mass fractions”. In fact, the definitions are given in liens 155; I suggest moving the definitions to lines 111.

Figures 1 and 2 left panels need more explanation. Sometimes only one number is given in the x-axis or y-axis legends; thus is not possible to understand the scale. Also, how can you explain % greater than 100 in figure 2f? Also, can you explain the meaning of the red numbers? Finally, the red curve is too thin and hard to be seen.

Linz 129: Have you an explanation to the fact the results are more accurate for the smallest particles than for the largest ones?

Line 144: Can you explain why temperature, pressure and humidity can affect the accuracy of the measurements?

Line 146: It is unclear how the particles are rejected. Is it by the instrument itself or by the machine learning method?

Author Response

Thank you for taking the time to read our manuscript. We really appreciate it!

All of the measurements were made by us, at our own field station, in the ambient environment using our own instruments. The calibration of the low cost AphaSense OPC occurs prior to their deployment across the dense urban environment of the Dallas Fort Worth Metroplex. In section 2 we have now added the following text to clarify this:

All of the measurements were made at our own field calibration station in the ambient environment. The calibration of the low cost AphaSense OPC occurs prior to their deployment across the dense urban environment of the Dallas Fort Worth Metroplex.

We completely agree that the type of particulates and their refractive index is an issue. With our current resources this is not something we can characterize. The machine learning methodology can easily be extended to cover this if we had the additional equipment to characterize the refractive index. Thanks for bringing this up, we appreciate it! We have added a new subsection 2.3.

2.3 Caveat: Particulate Refractive Index

The observations made by optical particle counters are sensitive to the refractive index of the particulates and their light absorbing properties. The retrieved size distributions and the mass-concentrations can be biased, depending on the nature of the particulates. The current study does not explore the accuracy implications of this. A future study is underway which includes direct measurements of black carbon that will allow us to begin to explore these aspects. The machine learning paradigm is readily extensible to include these aspects, even though not explicitly addressed in this study.

At your suggestion we have added a section to describe the operational use of this calibration and the post deployment ongoing calibration. A new section 5 has been added entitled Operational Use of the Calibration and Periodic Validation Updates.

5. Operational Use of the Calibration and Periodic Validation Updates}

The calibration just described occurs pre-deployment of the sensors into the dense urban environment. Once these initial field calibration measurements are made over a period of several months, in the manner described above, the multi-variate non-linear non-parametric empirical machine learning model is applied in real time to the live stream of observations coming from each of our air quality sensors deployed across the dense urban environment of the Dallas Fort Worth Metroplex. These corrected measurements are then made publicly available as Open Data as well as depicted on a live map and dashboard.

Building-in continual calibration to a network of sensors will enable long-term, consistent, and reliable data. While much effort has been recently placed on the connectivity of large disbursed IoT networks, little to no effort has been spent on the automated calibration, bias-detection, and uncertainty estimation necessary to make sure the information collected is sound. This is one of our primary goals. This is based on extensive previous work funded by NASA for satellite validation.

After deployment, a zero emission electric car carrying our reference is used, to routinely drive past all the deployed sensors to provide ongoing routine calibration and validation. An electric vehicle does not contribute any ambient emissions, and so, is an ideal mobile platform for our reference instruments.

For optimal performance, the implementation combines edge and cloud computing. Each sensor node takes a measurement at least every 10 seconds. The observations are continually time-stamped at the nodes and streamed to our cloud server, the central server aggregating all the data from the nodes, and managing them. To prevent data loss, the sensor nodes store any values that have not been transmitted to the cloud server for reasons, including communication interruptions, in a persistent buffer. The local buffer is emptied to the cloud server at the next available opportunity.

Data from all sensors are archived and serve as an open dataset that can be publicly accessed. The observed Probability Distribution Functions (PDFs) from each sensor are automatically compared in real time to the PDFs from the neighboring sensors within a neighborhood radius. These neighborhood sensors include measurements from the electric car/mobile validation sensors. This comparison is used to estimate the size resolved measurement uncertainty and size resolved inter-instrument bias for the last hour, day, week, month, and year. We continuously accumulate the PDF for each sensor over a variety of time scales (an hour, day, week, month, and year) and compare it to its nearest neighbors within a neighborhood radius.

Any calibration drift in a sensor will be quickly identified as part of a fully automated real-time workflow, where we will automatically be comparing each sensor’s PDFs to its neighbor’s PDFs, and to the reference instruments PDFs. As each sensor is in a slightly different local environment, the sensor bias drift for each sensor will be different. We have previously shown that machine learning can be used to effectively correct these inter-sensor biases [79]. As a result, the overall distributed sensing system will not just be better characterized in terms of its uncertainty and bias, but also provide improved measurement stability over time.

Line 33: Replace “90” by “90°”.

Thanks we corrected this.

Line 34: Replace “120” by “120°”.

Thanks we corrected this.

Line 111: Can you define “standardized occupational health Repairable, Thoracic and Alveolic mass fractions”. In fact, the definitions are given in liens 155; I suggest moving the definitions to lines 111.

Thanks, great idea, we moved the text as suggested.

Figures 1 and 2 left panels need more explanation. Sometimes only one number is given in the x-axis or y-axis legends; thus is not possible to understand the scale. Also, how can you explain % greater than 100 in figure 2f? Also, can you explain the meaning of the red numbers? Finally, the red curve is too thin and hard to be seen.

Thanks, we have expanded the explanation as follows:

For the left hand column of plots in Figure 1 (the scatter diagrams), for a perfect calibration the scatter plot would be a straight line with a slope of one and a y-axis intercept of zero, the blue line shows the ideal response. We can see that multivariate non-linear non-parametric machine learning regression that we have used in this study employing an ensemble of decision trees with hyper-parameter optimization has performed very well (panels a, d and g). In each scatter diagram the green circles are the data used to train the ensemble of decision trees, the red pluses are the independent validation data used to test the generalization of the machine learning model.

For the middle column of plots in Figure 1 (the quantile-quantile plots), we are comparing the shape of the probability distribution (PDF) of all the PM abundance data collected by the expensive reference instrument to that of the the PM abundance provided by calibrating the low-cost instrument using machine learning. A log10 scale is used with a tick mark every decade. The dotted red line in each quantile-quantile plot shows the ideal response. The red numbers indicate the percentiles (0, 25, 50, 75, 100). If the quantile-quantile plot is a straight line that means both PDFs have exactly the same shape as we are plotting the percentiles of one PDF against the percentiles of the other PDF. Usually we would like to see a straight line at least between the 25th and 75th percentiles, in this case we have a straight line over the entire PDF, which demonstrates that the machine learning calibration has performed well.

Linz 129: Have you an explanation to the fact the results are more accurate for the smallest particles than for the largest ones?

Yes, this is addressed in the text as follows: We can see that the performance is best for the smaller particles that stay lofted in the air for a long period and do not rapidly sediment.

Line 144: Can you explain why temperature, pressure and humidity can affect the accuracy of the measurements?

The calibration needs to account for temperature and pressure and humidity because:

The total number density (as we know from the ideal gas equation PV=nRT) is a function of pressure and temperature. the OPC counts particles, and the total number density will be changing with P & T. Humidity changes will affect physical properties and size of the particles and needs to be accounted for.

Line 146: It is unclear how the particles are rejected. Is it by the instrument itself or by the machine learning method?

The reject ratio is the term used by the OPC hardware for when a particle is not totally in the beam, appropriately accounting for this is a key part of the machine learning calibration.

Thanks again for your time and effort in helping us improve the paper. It is greatly appreciated.

Reviewer 3 Report

I read very carefully the manuscript entitled "Using Machine Learning for Calibration of Airborne Particulate Sensors" by Wijeratne et al. The subject of the manuscript is of great interest and the expectations of the title are extremely high. However I must confess that the work does not have a great scientific value. On page 3 of the manuscript, line 86, it is written: "It is an empirical approach, so we do not need to provide a theoretical model". Frankly, I believe this statement is absolutely false for a scientific article: the objective of an article is not simply to report unverifiable data, but to make readers understand how a certain work has been carried out. Goal of science is progressing in knowledge: in this work there is no explanation on how machine learning has been applied to the problem, how thoracic or alveolar mass fractions have been derived ... there is no explanation for anything. It is only said that the authors have done some work (which they must not report because it is empirical and as such debatable ..... but because we do not know how it was performed we cannot understand if correct or not) and have reported the results.

My judgment is that this work has no scientific value even if the subject is of extreme interest. The article must be rejected in the present form. To be published, it must receive a global review, where a discussion of the models used and their validity must be included.

Author Response

Thanks for taking the time to read our paper, it is appreciated.

Thanks for carefully reading the paper. We have added subsections at the start of the paper outlining the relevance of this work. We have also expanded on the description of the machine learning models used. While we respect your views we do not share the same conclusion.

All the code has been made open source, all the data has been made open data, we are addressing a challenging issue, the calibration of low cost sensors. We clearly stated how the how the thoracic and alveolar mass fractions have been derived. The approach was rigorously validated, in fact beyond what is usually done for machine learning, and this validation was shown: Two scatter diagrams for each model were shown where the model is directly confronted with actual data. This includes an entirely independent validation dataset (shown in red on the scatter plots). Quantile-quantile plots are shown where we see that the empirical model also reproduces the shape of the probability distributions of the system. This is a stringent test. After three decades of research it is clear from painful first hand experience that we do NOT have a theory for everything. There is an important place for empirical models. All theoretical models are based in some way on data. Machine learning encapsulates the scientific method of rigorously testing models against independent data.

The conclusion has been expanded to include the wider relevance.

Round 2

Reviewer 2 Report

The authors have well considered and answered to my comments. Thus the paper can be accepted for publication.

Reviewer 3 Report

I read the manuscript "..." very carefully and the comments that the authors sent on the criticisms. I must very much appreciate the calm and politeness with which the authors responded and the great effort they made to modify the proposed text.
The new version is definitely much clearer and explains the procedure used well. My judgment is definitely positive: the text deserves to be published in the journal sensors and I'm sure it will have a great response in the scientific community